# Strawberry Flavor Is Influenced by the Air Temperature Differential during Fruit Development but Not Management Practices

**Anya Osatuke and Marvin Pritts \***

School of Integrative Plant Science, Cornell University, Ithaca, NY 14853, USA; aco56@cornell.edu
\* Correspondence: mpp3@cornell.edu

**Abstract:** The majority of cultivated strawberries (*Fragaria × ananassa*) in the northern United States (US) and Canadian provinces is grown in perennial matted rows across a range of soil types and microclimates. Management practices vary in fertilization rates, intensity of pesticide use, and the source of inputs depending on grower preferences. The objective of this study was to identify environmental and management factors that influence strawberry flavor attributes across a range of production systems. The cultivar Jewel was selected for its popularity in this region and reputation for excellent flavor. "Jewel" was sampled from regional farms and, concurrently, grown in a controlled field study with different inputs over three years. Soluble solids content (SSC) and titratable acidity (TA) across farms was found to be positively associated with the air temperature differential during fruit ripening. In controlled field studies, yield was correlated positively with total N in the form of synthetic urea, but not with the rate of applied organic nitrogen (N). Despite different levels of soil carbon inputs, N rates, pesticides, and microbial supplements, the fruit quality attributes, including SSC, TA, aromatic volatile concentration, and phenolics were not associated with treatment. A human sensory evaluation found no perceptible differences in flavor or aroma among contrasting treatments. Our study concludes that growers should invest in temperature management, rather than agricultural inputs, to influence SSC and TA of strawberry.

**Keywords:** *Fragaria × ananassa*; organic production; temperature; aroma volatiles; fruit quality; titratable acidity; soluble solids

## 1. Introduction

Taste tests have identified high soluble solids content (SSC) as the most desirable quality associated with strawberry flavor, followed by titratable acidity (TA) and the concentration of total volatile compounds [1,2]. In the northeastern United States (US), many growers sell directly to consumers, either by inviting customers to pick their own fruit or offering fruit at farm stands and farm markets. To be competitive against more convenient supermarket strawberries, growers select cultivars that produce flavorful, sweet strawberries that can be picked at peak ripeness to secure market share [3], and would likely alter cultural practices if this resulted in more highly-flavored fruit.

A few studies have reported differences in strawberry chemical composition in response to management practice. Organic management increased total sugar and ascorbic acid content in strawberry [4–6], whereas supra-optimal N (nitrogen) applications caused poor color, flavor, and reduced sweetness and firmness [7,8]. Cover crops have been found to cause a reduction in free soil N [9,10], so this could affect flavor and yield indirectly. Cover crops also can improve soil health [11], so using them and other carbon sources might improve flavor relative to fields without additional organic inputs. Some have speculated that improved soil health in organic fields could account for flavor differences in strawberries [6,12]. Studies examining the impacts of microbial supplements have found improvements in phenolic content amongst treated plants [13], and some studies have

found improved yields in association with mycorrhizal inoculation [14–16]. However, these factors have not been shown to consistently affect strawberry flavor attributes when controlling for cultivar and location and are not commonly implemented by growers specifically for improving flavor.

Strawberry fruits of the same cultivar may display a high degree of variability in quality characteristics from year to year. A negative correlation between SSC and post-bloom air temperature has been observed in strawberry [17]. One study observed two-fold increases in total fruit sugars in treatments from one year to the next, and another observed a 2°Bx change [18,19]. A greenhouse study found significant year-to-year variation in content of ellagic acid and ascorbic acid [20]. Within the same year, a 7 °C rise in average daily temperatures, from 15 to 22 °C towards the end of the growing season, was identified as a potential cause for reduced sweetness under subtropical growing conditions [21]. Fruits harvested from plants in a growth chamber at 15 °C for 3 weeks after flowering had higher SSC than fruit from plants grown at 22 °C. A negative correlation was found between SSC and mean temperature over the 8-day period before harvest across a 6-year span [22]. SSC in the day neutral cultivar Albion was found to be associated with warm temperatures early in fruit development but cooler temperatures closer to ripening [23]. One study found that average air temperature during ripening tended to be negatively associated with firmness and SSC, but that the magnitude of the response was different depending on cultivar [24]. Some cultivars were more stable than others, both within and between years. Clearly, weather patterns, and specifically temperature, influence sugars and acids in harvested fruit, but the specific time during fruit development when temperature is impactful has not been well quantified.

Northern US growers typically grow short day strawberries in a perennial matted row system because of better plant establishment and less risk of crop loss due to frost damage that might otherwise occur on plastic–covered beds [3]. The goal of our study was to take a holistic look at a suite of potential factors that might affect flavor in perennial strawberry. We measured flavor attributes across several farms over several years for a single cultivar, compared the variation from year-to-year with variation from farm-to-farm, and attempted to find weather variables that were associated with variation in flavor attributes. During those same 3 years, we grew the same cultivar using different sets of inputs (fertilizers, pesticides, microbial supplements, cover crops) and field management options available to northern US growers. Our hypothesis was that fruit quality and yield would be impacted by these different sources and rates of inputs.

## 2. Materials and Methods

### 2.1. Farm Survey Methods

Strawberry growers from across New York State (US) were recruited to participate in the study. The cv. Jewel was selected for its excellent flavor and popularity amongst northeastern US growers [3,25]. Fruit samples were collected at the peak of ripeness, once per year, from 22 farms over 3 years (2018–2020), although not all farms were sampled each year due to the travel distances involved and the short harvest season. Samples were visually assessed for ripeness and hand–picked. Samples were then placed into cardboard pulp containers (0.95 L), and the containers were inserted into 8 L freezer bags. Bags were kept in a closed, insulated cooler filled with ice packs and driven to Ithaca, NY where samples were cleaned of their calyx, rinsed with tap water to remove dirt and residues, and frozen to −10 °C within 24 h of harvest. Samples were then analyzed for SSC, titratable acidity (TA), and selected aromatic volatiles as described below. Harvest dates varied depending on the farm location.

Weather data were obtained from the Network for Environmental and Weather Applications (NEWA, 2020) for 2018–2020 [26]. For each farm, the nearest weather station was located using map software. Growing degree days (GDD base 15 °C) and cumulative rainfall was calculated from weather station records ranging from 1 April to 30 June at the site closest to each farm. These dates were selected because they spanned the time

from flower bud emergence to fruit ripening. Cumulative solar radiation, mean daily temperature, highest daily temperature, mean low temperature, and lowest minimum temperature were calculated for four discrete fruit developmental stages: April 1 until flowering (Stage 0, variable length), flowering until green fruit (Stage 1, 15 days), green fruit until pink (Stage 2, 10 days), and pink fruit until ripe fruit (Stage 3, 5 days). Dates for the developmental stage of each harvest were back–calculated from the date that the fruit was harvested, based on the assumption that fruits were ripe 30 days after flowering. Correlations were made between fruit quality variables and weather variables during each stage, including the temperature differential between average air temperature and minimum low temperature. Data were not collected on soil type or source of planting stock for farms in the sample.

### 2.2. Research Farm Methods

A study was conducted at a research farm in Ithaca, NY concurrently with the regional farm survey. A field of Arkport fine sandy loam (mesic Lamellic Hapludalf) at the Cornell University Agricultural Experiment Station (42.441582, −76.472487) was divided into 72 plots that were 4.6 × 3.6 m. Eight replications of nine treatments were randomly assigned to plots. Treatments represented a range of management practices that are commonly used for perennial matted row strawberry production in New York State (Table 1). The nine treatments included three different input sources: Organic (ORG), Conventional (CON), and Low Carbon (LOC). Each input source had three levels of intensity: (−) treatments received 50 kg N·ha$^{-1}$ annually, (o) treatments received 100 kg N·ha$^{-1}$, and (+) treatments received 100 kg N·ha$^{-1}$ plus additional applications of non–fertilizer inputs. A standard grower recommendation in this region is 100 kg N·ha$^{-1}$ annually [27]. A cover crop (winter rye, *Secale cereale* L.) was seeded in 48 of the 72 randomly assigned plots (CON and ORG treatments) in October 2017 and incorporated in May 2018 prior to planting strawberries. Dormant "Jewel" strawberries were planted on 14 May 2018. Rows were 1.2 m on center with plants set 0.45 m apart. The two center rows of each plot served as data rows, and the outer rows served as shared treatment borders with adjacent treatments.

LOC+ and CON+ treatments received applications of herbicides at standard rates (terbacil at 92 g ha$^{-1}$) at monthly intervals during the growing season, napropamide (8 kg ha$^{-1}$) post–plant and in the fall, and fungicides (captan at 92g ha$^{-1}$) at recommended rates [27]. Other LOC and CON plots were treated only with napropamide post-planting at the same rate as (+) treatments. ORG+ received a pre-plant root dip (MycoGold® 1% for 1.5 h, Amelia, OH) and monthly applications of Growcentia foliar microbial biostimulant (Yeti® at a rate of 2.65 mL L$^{-1}$, Fort Collins, CO, USA) applied to runoff during the growing season. For CON and ORG treatments, straw mulch was applied between rows after establishment, and annually over top of rows prior to winter for cold temperature protection. Straw mulch and cover crop were withheld from all LOC treatments so floating row cover was used for winter protection. LOC treatments were temporarily mulched with black plastic between rows for 3 weeks during the fruiting period to keep berries clean. These treatments created a range of N, organic matter, and crop microbial supplementation involving either conventional or organic products (Table 1). Insecticides were not used in this study.

Strawberry harvest from field plots occurred on 19, 26, 28 June, and 2 and 10 July 2019, and 17 and 23 June 2020 after almost 2–3 years under the various treatments. In 2020, several unusual late frosts in May during flowering diminished early yield potential and reduced the number of harvests. Fruit samples were collected from 1–m sections in each of two rows near the center of each plot, representing 27–31 plants. Border rows were excluded from harvest. Cumulative yield, percent marketable fruit, and average individual fruit weight were recorded for each plot. Individual fruits were categorized as unmarketable if they had sun damage, malformation of the fruit tip, holes from bird pecks or insects, soft spots indicative of early fungal infection, mold, fruit size smaller than 5 g, or irregularly shaped fruit. Fruit within central rows but outside of the internally-marked

data plot were analyzed for firmness after harvest on 23 June, 30 June, and 9 July, with materials and methods as indicated by Shin et al. (2008) [28].

**Table 1.** Perennial matted row treatments for "Jewel" strawberry between Fall, 2017 and Spring, 2020 in Ithaca, NY. N = nitrogen. ORG = organic inputs, CON = conventional inputs, and LOC = low carbon inputs. (−) = low nitrogen fertilizer, (o) = recommended rate of N fertilizer, (+) = recommended rate of N fertilizer plus supplemental inputs.

| Treatment | Annual N Rate | N Source | Organic Matter Supplementation | Microbial Supplementation | Synthetic Pesticide Use |
|---|---|---|---|---|---|
| ORG − | 50 kg N·ha$^{-1}$ | Compost, composted chicken manure | Straw mulch, *Secale cereale* preplant cover crop | None | None |
| ORG o | 100 kg N·ha$^{-1}$ | Compost, composted chicken manure | Straw mulch, *Secale cereale* preplant cover crop | None | None |
| ORG + | 100 kg N·ha$^{-1}$ | Compost, composted chicken manure | Straw mulch, *Secale cereale* preplant cover crop | MycoGold® pre-plant, Yeti® at a rate of 10 mL·3.78 L$^{-1}$ monthly | None |
| CON − | 50 kg N·ha$^{-1}$ | Urea | Straw mulch, *Secale cereale* preplant cover crop | None | napropamide, 8 kg·ha$^{-1}$ pre-plant annually |
| CON o | 100 kg N·ha$^{-1}$ | Urea | Straw mulch, *Secale cereale* preplant cover crop | None | napropamide, 8 kg·ha$^{-1}$ pre-plant annually |
| CON + | 100 kg N·ha$^{-1}$ | Urea | Straw mulch, *Secale cereale* preplant cover crop | None | napropamide, 8 kg·ha$^{-1}$ pre-plant annually, captan, 92 g·ha$^{-1}$ in spring, terbacil, 92 g·ha$^{-1}$ monthly |
| LOC − | 50 kg N·ha$^{-1}$ | Urea | None | None | napropamide, 8 kg·ha$^{-1}$ pre-plant annually |
| LOC o | 100 kg N·ha$^{-1}$ | Urea | None | None | napropamide, 8 kg·ha$^{-1}$ pre-plant annually |
| LOC + | 100 kg N·ha$^{-1}$ | Urea | None | None | napropamide, 8 kg·ha$^{-1}$ pre-plant annually, captan, 92 g·ha$^{-1}$ in spring, terbacil, 92 g·ha$^{-1}$ monthly |

Leaf samples were collected from center rows on 7 Sept 2019. Total N and carbon (C) was analyzed by Cornell Nutrient Analysis Laboratory using an elemental analyzer (NC2500, Carlo Erba, Milan, Italy) coupled to an isotope ratio mass spectrometer (Delta V, Thermo Scientific) [29].

Odor and flavor assessments of research farm samples were conducted to determine if human subjects could distinguish among various treatments. The same-different testing method was used [30]. The flavor assessment was conducted on 11 July 2019 using berries harvested on 10 July 2019. Strawberries were thoroughly rinsed in tap water prior to presenting berries to panelists. To prepare a pair of samples, strawberries had the calyx removed and were then weighed. To control for latent bias for size, strawberries with weight under 5 g were paired with a strawberry weighing within 1 g of the first, berries with a weight between 5 to 10 g were paired with a berry within 2.5 g, and berries with weight over 10 g were paired with a berry within 5 g of the other. Ripeness and fruit quality were controlled during harvesting by only using fully ripe marketable berries. Panelists

received optional saltine crackers and plain seltzer water to clear their palate between tastings. Panelists were asked to compare six treatments and indicate if the samples were the same or different, with sample pairs presented in randomized order. Twenty-eight panelists participated in the flavor assessment.

Odor assessment was conducted on 26 June 2019 using fruit harvested on 25 June 2019. Strawberries were halved using a knife, and half of each strawberry was placed in a 59 mL plastic Solo cup, then immediately capped in foil for 0 to 5 min prior to delivery to the sensory panelist. Immediately before giving a pair of samples to the panelist, a fork was used to create holes in the foil cover of the cup. Panelists were asked if the odors were the same or different. Panelists were then provided coffee beans to smell to prevent nasal fatigue by necessitating the act of sniffing between samples [31,32]. Each panelist was asked to evaluate 4 paired sample comparisons, and each pairing was immediately followed by a repetition of the same pairing in a randomized order. Thus, each panelist assessed a total of 16 berries. Panelists were asked to compare six treatments, with sample pairs presented in randomized order. Twenty-nine panelists participated in the odor assessment.

For sensory analyses, the likelihood of panelists correctly identifying a pair of different berries was derived from the fractional odds ratio using the following equation: Odds of correctly identifying (ID) a different pair: [(Correct ID of different pairs/(Correct ID of same pairs + Correct ID of different pairs)]/[(Incorrect ID of different pairs/(Correct ID of same pairs + Correct ID of different pairs)]. Odds were divided by (odds + 1) and multiplied by 100% to calculate probabilities.

Fruits collected on 2 July 2019 were analyzed for phenolic content. Phenolic content was evaluated on a subset of treatments using Fast Blue BB (D 45–44670–50G–EA, Krackeler Scientific, Albany, NY, USA) assay [33]. Each field plot was subsampled for fruit thrice, yielding three juice replicates per plot. Analyte was pipetted into 96–well plate (655185, VWR International, Randor, PA, USA). Juice samples were compared against gallic acid standard solutions with a concentration of 0, 0.01562, 0.02135, 0.0625, 0.125, and 0.50 mg mL$^{-1}$, and against ascorbic acid standard dilutions of 0, 0.01562, 0.02135, 0.0625, 0.125, 0.50, and 1.0 mg mL$^{-1}$. Absorbance was measured on SoftMaxPro 7.0 (2016–2020, Molecular Devices, San Jose, CA, USA) at 420 nm. The calibration curve was linear in the range studied with a correlation coefficient of 0.98.

### 2.3. Methods for Both Regional Farm Samples and Research Farm Plots

For both grower and research farm samples, SSC and TA were measured using an ATAGO 7104 PAL–BX/ACID4 Pocket Titrimeter-Refractometer for Strawberry (ATAGO USA, Bellevue, WA, USA). Values were calculated from a 3–berry subsample. Frozen samples were thawed for 2h and juice was squeezed for analysis. The device calculated SSC through refractometry, and TA of 1:50 juice:water dilution through a built–in conductivity probe. Acid correction for SSC was calculated using the following equation: Corrected SSC (%) = Original SSC(%) − (TA(%) × 0.1943), but the corrections were small and did not affect the results; therefore, uncorrected SSC values were used in analyses [34].

Samples for volatile analysis were prepared using methods from Jetti et al. (2007) [35], and frozen in a −80 °C freezer prior to analysis. Samples were thawed for 90 to 120 min at room temperature prior to loading into the injection port. Prior to statistical analyses, target volatiles were first identified among the hundreds that exist using samples collected in 2018 [36]. Major volatiles associated with flavor in "Jewel" included furanone, isoamyl acetate, hexanoic acid, ethyl butanoate, ethyl hexanoate, and linalool (Table 2), so only these were used in further analyses.

Volatile quantification of fruit samples from both grower farms and the research farm was performed using a gas chromatograph (GC) coupled to a triple quadrupole mass spectrometer (Shimadzu, TQ8040, Tokyo, Japan). A 2-cm divinylbenzene/Carboxen/ polydimethylsiloxane (DVB/CAR/PDMS) fiber (SU57299U, Neta Scientific, Hainesport, NJ, USA) fitted to a CTC CombiPal (Agilent, Santa Clara, CA, USA) autosampler was used for solid-phase microextraction (SPME), with an incubation temperature of 37 °C for 15 min and agitated at 500 RPM.

The SPME fiber was incubated in the sample headspace for 60 min at 37 °C and desorbed in a splitless injection port for 3 min at a constant temperature of 230 °C.

**Table 2.** Mean value of normalized peak area (cumulative sum of ion intensity/min retention time) and coefficient of variation (CV) of six volatiles measured in "Jewel" strawberry fruit collected in 2018 from 22 farms across New York State.

| Compound | Normalized Peak Area | CV |
|----------|:--------------------:|:----:|
| Furanone | 0.4 | 119% |
| Isoamyl acetate | 0.2 | 122% |
| Ethyl hexanoate | 0.6 | 83% |
| Linalool | 2.0 | 86% |
| Hexanoic acid | 1.5 | 100% |
| Ethyl butanoate | 0.6 | 93% |

Injections were splitless with a flow rate of 1.79 mL/min$^{-1}$ purged after 1 min. Ultra high purity helium was used as the carrier gas with a flow rate of 1 mL/min$^{-1}$. The gas chromatograph was equipped with a Varian FactorFour™ capillary column (30 m × 0.25 mm × 0.25 µm, CP9205, Walnut Creek, CA, USA). Initially, the GC oven was held at 25 °C for 5 min, then ramped to 40 °C at a rate of 5 °C/min, and held at 250 °C for 10 min. The MS was operated in EI mode with an ionization energy of 70 eV in SIM mode with an ion source temperature of 230 °C and an interface temperature of 230 °C. Peak areas of target volatiles were quantified using Shimadzu GCMSolutions software using the following quantifier ions: 3-heptanone [*m/z* 57, 85, 41], naphthalene D8 [*m/z* 1136, 108, 134], and isopropyl methoxy pyrazine d3 [*m/z* 139, 121, 81].

R-software was used for all statistical analyses, (Version 1.1.456, 2009–2018 RStudio, Inc., Boston, MA, USA). For farm samples, relationships between flavor attributes and GDD, cumulative solar radiation, cumulative rainfall, and temperature variables (average air temperature, minimum air temperature, and the air temperature differential) for each of the four stages of fruit development were evaluated using linear regression. Year-to-year and farm-to-farm differences were assessed using data from the 8 farms visited in 2018, 2019, and 2020. ANOVA was used to determine if statistical differences existed for SSC and TA between years when farm served as a replicate, and conversely, if differences existed between farms when year served as a replicate. Pearson's correlation coefficients and *p*-values for the slope of the linear equations were calculated. For the field study, data were analyzed as a completely randomized design using ANOVA. Tukey's HSD with a significance level $\alpha = 0.05$ was used to determine differences among significant factors.

## 3. Results

### 3.1. Grower Samples

We examined many different air temperature patterns during 4 fruit developmental stages prior to fruit ripening across the regional farms. We found strong correlations between mean air temperatures during the green to pink fruit ($r = -0.39$, $p < 0.006$) and pink to ripe fruit stages ($r = -0.36$, $p < 0.03$; Figure 1), but the lowest minimum air temperature also was negatively correlated with SSC ($r = -0.42$, $p < 0.003$; Figure 2) between first flower and green fruit. Therefore, we subtracted the lowest minimum air temperature from the mean air temperature to generate the temperature differential for each stage of fruit development. This differential was even more strongly associated with SSC for both Stage 1 and Stage 2 ($r = 0.43$, 0.36; $p < 0.03$, 0.01 respectively, Figure 3). A similarly strong relationship was found with TA ($r = 0.75$, 0.74; $p < 0.0001$, respectively; Figure 4). SSC and TA were strongly correlated with each other in our study ($r = 0.45$, $p < 0.01$) so their relationship with the temperature differential was similar. Positive correlations also were found between solar radiation and TA at each stage of fruit development, but correlations with the temperature differential were stronger than correlations with any other weather variable.

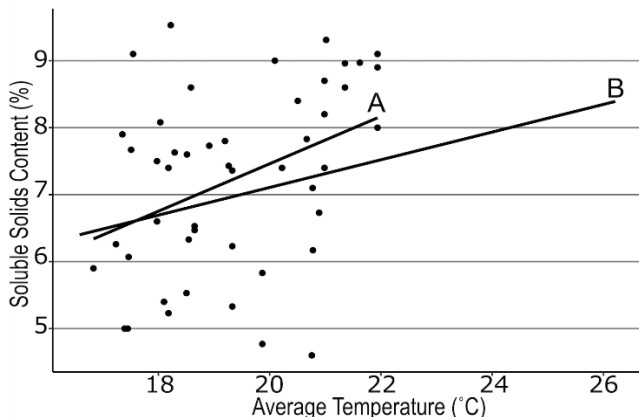

**Figure 1.** Regression between soluble solids content of strawberry fruit at harvest and the mean temperature between green and pink fruit (line A) and pink and red fruit (line B) for 16 farms across New York State visited in 2018, 2019, and 2020. A: y = 0.35x + 0.4, *r* = 0.39, *p* = 0.0062. B: y = 0.21x + 3.0, *r* = 0.36, *p* = 0.03.

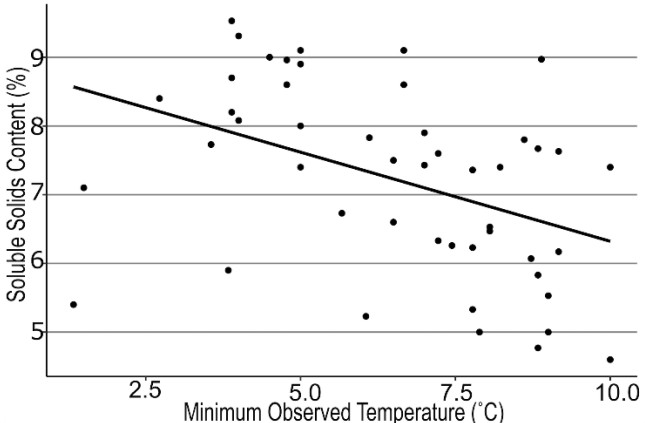

**Figure 2.** Regression between soluble solids content of strawberry fruit at harvest and the lowest single temperature between first flower and green fruit for 16 farms across New York State visited in 2018, 2019, and 2020. y = −0.26x + 8.9, *r* = −0.42, *p* < 0.0027.

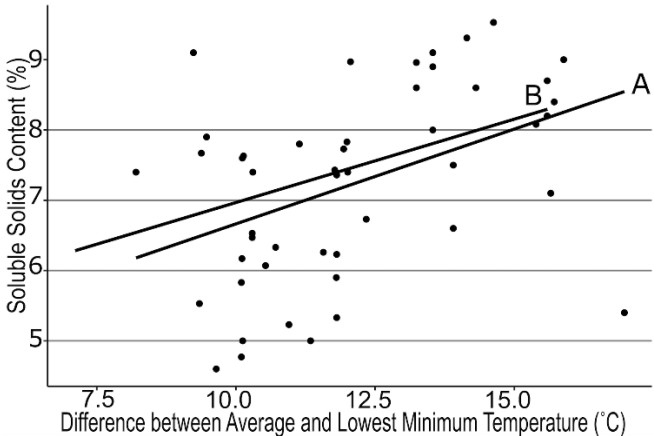

**Figure 3.** Regression between final soluble solids content of strawberry fruit at harvest and the difference between mean air temperature and the lowest single temperature between first flower and green fruit (A) and green to pink fruit (B) for 16 farms across New York State visited in 2018, 2019, and 2020. A: y = 0.27x + 4.0, *r* = 0.43, *p*< 0.005. B: y = 0.24x + 4.6, *r* = 0.36, *p* < 0.05.

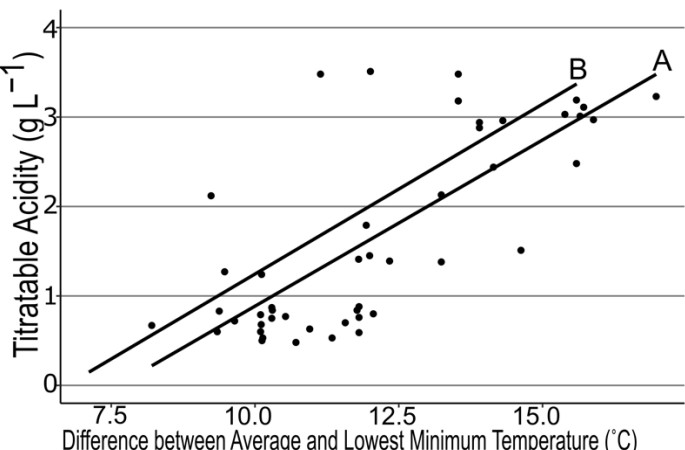

**Figure 4.** Regression between titratable acidity of strawberry fruit at harvest and the difference between mean air temperature and the lowest single temperature between first flower and green fruit (A) and green to pink fruit (B) for 16 farms across New York State visited in 2018, 2019, and 2020. A: y = 0.37x − 2.8, r = 0.7, p < 0.0001. B: y = 0.38x − 2.6, r = 0.74, p < 0.0001.

The relative importance of year-to-year variation and farm–to-farm variation was determined by examining the coefficients of variation (CV) of the measured variables. The CV of the population of differences from one year to the next on the same farm (70–360% for SSC; 34–49% for TA) was much larger than the CV of the population of differences among farms in the same year (12–18% for SSC; 11–25% for TA). Furthermore, SSC and TA levels increased or decreased in a consistent direction from one year to the next across all farms (Figures 5 and 6). Accumulated GDD and rainfall from 1 April were either weakly or not consistently correlated with SSC or TA. SSC and TA were significantly different between years when treating farm site as a replicate ($p < 0.05$, <0.0001, respectively), but the effect of farm site was not significant when treating year as a replicate ($p > 0.3$, $p > 0.9$, respectively).

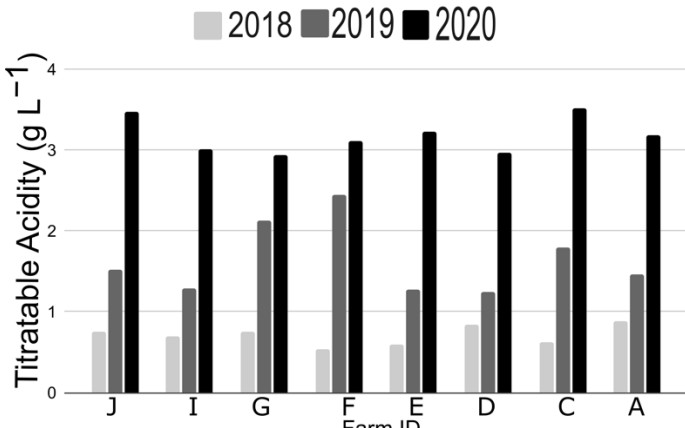

**Figure 5.** Titratable acidity of "Jewel" between 2018, 2019, and 2020. Each letter corresponds to an individual farm in New York State. One sample was collected per farm between early June and early July. Differences between years were significant ($p < 0.0001$) when treating farms as replicates, but not between farms when treating year as a replicate.

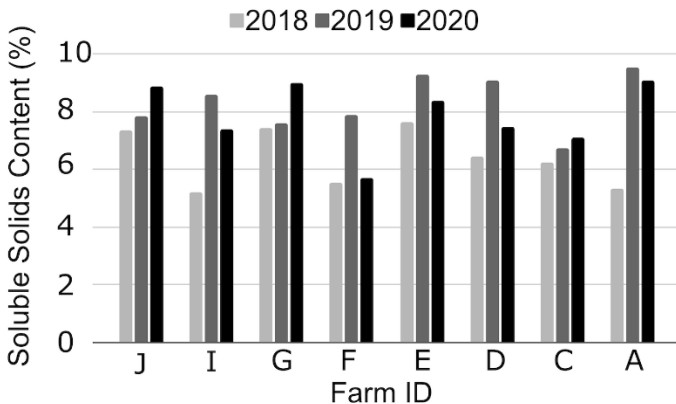

**Figure 6.** Soluble solids content (%) of "Jewel" strawberry in 2018, 2019, and 2020. Each letter corresponds to an individual farm in New York State. One sample was collected per farm between early June and early July. Differences between years were significant ($p < 0.05$) when treating farms as replicates, but not between farms when treating year as a replicate.

Variation was observed in volatile content from farm-to-farm as well (Table 2), but this variation was not associated with temperature patterns, solar radiation or rainfall. All volatiles exhibited a large amount of variation among the fruit samples (Figure 7), and volatile levels tended not to be correlated with TA or SSC within a given year. Neither farm location nor the weather variables we measured were influencing aroma volatiles in a consistent way.

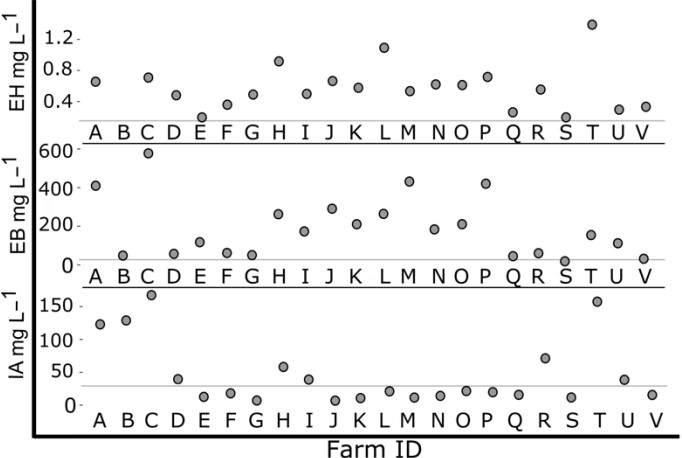

**Figure 7.** Estimated concentrations of three esters in "Jewel" across regional farms in New York State. Esters: ethyl hexanoate (EH), ethyl butanoate (EB), isoamyl acetate (IA). Fruits collected between 6 June and 3 July 2018. Gray horizontal line indicates threshold for ester flavor detection: Nasal threshold of ethyl hexanoate at 0.16 m%, orthonasal threshold of ethyl butanoate at 20 m%, and orthonasal threshold of isoamyl acetate at 30 m%.

### 3.2. Research Farm Trials

Among all of the response variables that we measured, only yield and % leaf N were affected by treatment. All ORG treatments had approximately the same value for leaf N (1.84%), regardless of the rate of N application. At the 100 kg N·ha$^{-1}$ rate, organically-managed treatments (ORGo and ORG+) had leaf N values about one half percent lower than leaves from conventionally-managed plants fertilized with the same rate of N. Leaf N increased significantly with N application in LOC treatments and the same trend existed for CON treatments. There were no significant differences in leaf C concentration among treatments, despite withholding organic amendments (cover crops, straw) from the LOC set

of treatments. The leaf C/N ratio was significantly higher in all ORG treatments compared to conventionally-managed treatments (Table 3).

**Table 3.** Mean values for leaf nitrogen (N), and carbon/nitrogen (C/N) ratio among strawberries grown under nine different management regimes. ORG = organic inputs, CON = conventional inputs, LOC = low carbon inputs; (−) = low nitrogen fertilizer, (o) = recommended rate of N fertilizer, (+) = recommended rate of N fertilizer plus supplemental inputs.

| Treatment | Leaf N | Leaf C/N Ratio |
|:---:|:---:|:---:|
| ORG − | 1.84% c | 26.2 a |
| ORG o | 1.84% c | 26.1 a |
| ORG + | 1.84% c | 26.4 a |
| CON − | 2.11% abc | 22.7 bc |
| CON o | 2.33% a | 20.6 c |
| CON + | 2.30% a | 20.9 c |
| LOC − | 1.96% bc | 24.8 ab |
| LOC o | 2.21% ab | 21.6 bc |
| LOC + | 2.26% a | 21.3 c |

Values followed by different letters within a column are significantly different at $p < 0.05$ using Tukey's HSD.

In 2019, the average yield across all treatments was 2.91 kg·m$^{-2}$ of row, ranging from 1.72 to 4.1 kg·m$^{-2}$ (12.5–30 t·ha$^{-1}$ equivalent). When fertilized at the maximum rate of 100 kg N·ha$^{-1}$, CON+ and LOC treatments yielded significantly more total and marketable fruit than all other treatments, including ORG+. ORG+ treatments yielded 21% less total yield than CON+, and ORGo yielded 25% less (Figure 8). No significant differences were observed in the percentage of yield that was marketable between treatments (mean of 68%). In 2020, yield was low, and treatment differences were not significant as several unusual late spring frosts in early May diminished yield potential. Yield declined from 2.91 kg·m$^{-2}$ in 2019 to 0.67 kg·m$^{-2}$ in 2020.

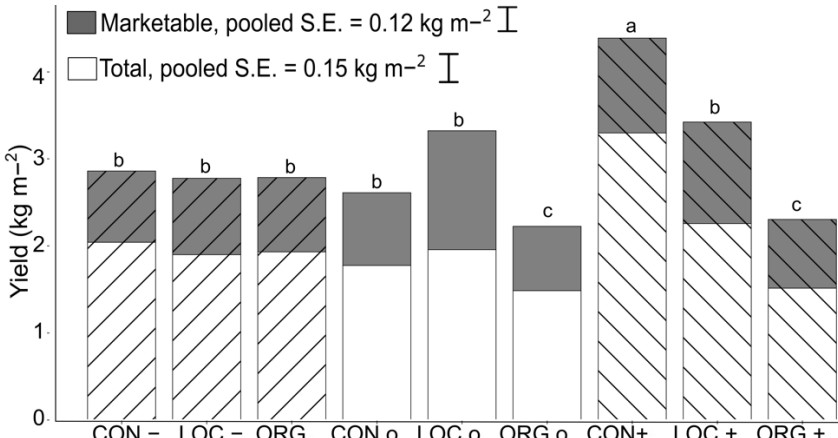

**Figure 8.** Total yield and marketable yield of "Jewel" strawberry for various management practices. Fruits were collected between 19 June and 10 July 2019. Management regimes: Conventional (CON), Organic (ORG), and Low Carbon Conventional (LOC). Treatments fertilized with 50 kg N ha$^{-1}$ (−), cross hatch up to right, 100 kg ha$^{-1}$ N (o), no cross hatch, or 100 kg N ha$^{-1}$ with intensive application of biological supplements (+), cross hatch down to right. Differences between treatments calculated using Tukey's HSD, where differences in marketable yield were significant at $p = 0.094$, and differences in total yield significant at $p = 0.05$. Letters indicating significance are identical for both total and marketable yields. S.E. = standard error of the mean.

No significant differences were observed in average fruit weight between treatments (12.3 +/− 0.66 g) in 2019. Average individual fruit weight significantly decreased over the course of the harvest season ($p < 0.001$), but did so consistently across all treatments.

In 2020, average individual fruit weight decreased by 1.5 g between the first and second harvest (7.45 vs. 5.95 g).

There were no significant differences in firmness (10.3 +/− 0.27 N) between treatments at any single date in 2019, although there was a tendency of firmness to decrease with higher N rates in the form of urea. Firmness significantly decreased over the harvest season, with most fruits decreasing by approximately 3.17 N each week ($r = -0.81$, $p < 0.0001$).

There were no significant differences between treatments in SSC (5.95 +/− 0.12%), nor was there a change in SSC over the course of the 2019 season. Again, in 2020, no significant differences were observed in SSC among treatments (8.01 +/− 0.16%), although SSC was much higher than in 2019. In 2019, there were no significant differences in TA between treatments (1.41 +/− 0.05%), but TA increased by approximately 0.12% with each consecutive harvest. Total phenolic content was not significantly different between samples from CON +, ORG +, LOC +, and LOC− treatments, with a mean gallic acid equivalent measurement of 1.06 +/− 0.04 mg·mL$^{-1}$. Volatile content amongst field treatments exhibited extremely high variation (CV ranged from 90 to 176%), but this could not be associated with any imposed management practice.

Panelists were unable to distinguish differences between selected contrasting treatments by either smell or taste (Table 4). They were as likely to say that two samples were the same when they were different as to say samples were different when they were the same. Correct responses never exceeded 50% for any set of comparisons, indicating a random pattern of discernment.

**Table 4.** Likelihood of correctly identifying a different pair of "Jewel" strawberries grown under different management regimes in two separate sensory evaluations. The taste test compiles observations from 28 panelists, the sniff test compiles observations from 29 panelists. Management regimes: Conventional (CON), Organic (ORG), and Low Carbon Conventional (LOC). Treatments fertilized annually with 50 kg N·ha$^{-1}$ (−), 100 kg N·ha$^{-1}$ (o), or 100 kg N·ha$^{-1}$ with intensive application of biological supplements (+).

| Plot Comparison | Probability of Correctly Identifying a Difference in Taste Test | Probability of Correctly Identifying a Difference in Sniff Test |
| --- | --- | --- |
| CON + to ORG + | 32% | 31% |
| CON o to LOC + | 20% | 41% |
| CON o to ORG o | 17% | 28% |
| CON o to CON − | 15% | 32% |

## 4. Discussion

Strawberry quality attributes from regional farm samples measured as SSC and TA were strongly influenced by the year the samples were taken. Nearly all measurements changed in the same direction from one year to the next. SSC was lowest in 2018 (6.32%) and highest in 2019 (8.32%). Every farm exhibited increased SSC and TA in 2019 compared to 2018, and fruit from all farms increased in TA from 2018 to 2019 to 2020. This consistent pattern suggests that regional weather may have been controlling TA and SSC at harvest. Indeed, when we closely examined weather data before harvest, we found that the differential between the average daily temperature and the lowest minimum temperature between flowering to pink fruit was most strongly associated with high SSC and TA content in ripe fruit, more so than average daily temperature or lowest minimum temperature alone. Solar radiation was less strongly correlated with SSC and TA, but this variable is collinear with the temperature differential as clear skies are associated with both warm days and cool nights. Rainfall was not significantly correlated with TA or SSC across years or within a year, likely because growers manage soil moisture with irrigation, so the commercial plantings were never affected by lack of rainfall.

Our findings suggest that warm days coupled with occasionally cold nights promoted sugar accumulation in developing fruit. Many other studies have found that exposing



developing fruit to high temperatures decreases SSC content in ripe fruit. For example, in temperate climates, high air temperatures have been found to negatively correlate with SSC [17]. Subtropical production areas see a rise in temperatures over the course of the season, and the seasonal decline in sweetness is relatively well-documented [22,24,37]. Both warm and cool temperatures can be associated with SSC and TA accumulation, although their positive effects occur at different times during fruit development. The closer to ripening, the more warm temperatures negatively affect SSC at harvest [23]. Our study found that the larger the difference between warm and cool temperatures during fruit development prior to coloring, the more SSC and TA at harvest. In contrast to temperature effects on SSC and TA, we did not find a relationship between weather variables or the temperature differential with aroma volatiles. Volatile amounts were extremely variable from year-to-year and from farm–to-farm, and we were unable to discern any pattern.

Year-to-year variation in SSC and TA content during fruit development is large enough to have perceptible impacts on flavor perception and is likely a major contributor to the reputation for inconsistency in strawberry quality [1]. A trained sensory panel can detect a perceptible change in sour taste with a 0.08% shift in TA, while a perceptible change in sweet taste can be expected at 1% change, although SSC was found to be a less dependable predictor for sweet taste than the TA measurement for sour taste [38]. The cited research was conducted using apples, a fruit with >90% of acidity coming from malic acid [39]. Conversely, the dominant acids in strawberry are citric and malic acid [40]. The perceived sourness in a food is not solely dependent on titratable acidity, but at equivalent concentrations, malic acid evokes a greater sour taste than citric acid in humans. Citric acid is estimated to be approximately 70–85% as tart as malic acid [41]. Extrapolating from these findings, a shift in TA of 0.092–0.104% may be required for a detectable difference in sourness in strawberry fruit. We found differences in SSC between regional farms in the same year of 1.5% and 0.15% for TA. This is above the threshold for detectable differences (Figures 5 and 6), so would allow consumers to recognize flavor differences with the same cultivar among farms as well as between years. Our data suggest that differences among farms may not be statistically significant, although our sample size was small.

Our field trial compared various management practices on the same farm, so weather and initial soil variables were identical for all treatments. In 2019, yield was significantly impacted by N rate and form. All ORG treatments in the study were deficient in N and had the same % leaf N regardless of the fertilization rate. Even treatments fertilized with 100 kg N·ha$^{-1}$, the recommended rate for New York perennial strawberries, did not supply sufficient N in an organic form. ORG plots received the first N supplementation earlier than CON plots in the form of incorporated compost and afterwards were supplemented with composted chicken manure on the same dates as CON plots. This suggests that the N in the manure was not readily available to the strawberry plants in sufficient quantities for good growth and yield. In poultry manure, approximately 55% of N will be released in the season of application, and approximately 12% more N will be available to the plants the following year [42]. In contrast, much of the N in the form of urea can be taken up within the season of application, barring losses to leaching or volatilization. Plants responded in predictable ways to urea with higher % leaf N and increased yields at the higher N rate. However, we were not able to identify differences in fruit chemical composition that could be attributed to N rate or form. In spite of these differences in vegetative and yield response to organic and conventional N sources, our sensory panelists were unable to detect differences in flavor associated with N source or rate.

Neither leaf N nor total or marketable yield in CON treatments with incorporated rye and straw mulch were significantly different than in treatments without these organic matter amendments (LOC), suggesting that these C inputs did not affect N availability. Although yield was not affected by the addition of straw mulch in our study, berries had less soil on their surface in straw-mulched treatments. One study found that the use of straw mulch increased strawberry sugar content [16], but we did not observe this effect. In subtropical growing regions, the use of black plastic mulch has been found to significantly

reduce yields by compounding heat stress [43,44], so mulch choice can be impactful in some situations.

Three of the nine treatments received additional non–fertilizer inputs. LOC+ and CON+ received frequent applications of synthetic pesticides, and plants with ORG+ were given a pre-plant root dip and monthly foliar applications of a microbial supplement during the growing season. The treatment with conventional supplements (CON+) had greater total and marketable yields than the equivalent treatment without supplements, likely because the monthly applications of terbacil herbicide suppressed weeds better than a single spring application of napropamide (CONo). Organic plots (ORG −, o, +) required more labor for hand–weeding, and yields were lower even with supplementation. None of the supplemental applications affected flavor attributes.

## 5. Conclusions

Our study suggests that differences in common management practices contribute little to flavor variation in short-day perennial strawberries. SSC and TA content were not significantly different among fruit from diverse treatments, and taste panelists were unable to distinguish fruit from them. Plant N content and yield increased with the amount of applied N, but only when the N was in the form of urea rather than composted manure. Yields were highest with 100 kg N·ha$^{-1}$ combined with monthly applications of herbicide, a practice that is commonly used by conventional growers. Our research suggests that in perennial matted row production systems, yield does not appear to be negatively correlated with flavor attributes. Rather, we found that fruit sugar and acid levels are associated with warm days and an occasional cold night between flowering and pink fruit. Aromatic volatiles were not strongly affected by weather. The claim that certain management practices or input sources strongly impact flavor is not supported by our data. Our study used only one cultivar. Other cultivars may be more responsive to management practices and input sources.

**Author Contributions:** Conceptualization, methodology, formal analysis, resources, validation, review and editing, A.O. and M.P.; data curation, software, investigation and original draft preparation, A.O.; resources, funding acquisition, project management, project administration and visualization, M.P. All authors have read and agreed to the published version of the manuscript.

**Funding:** Funding for this research was provided by the North American Strawberry Growers Association, New York State Agriculture and Markets and the National Institute of Food and Agriculture, U.S. Department of Agriculture, Hatch grant 1017299.

**Institutional Review Board Statement:** Ethical review and approval were waived for this study, based on subsection 6i of Section 46.104 of the 2018 Common Rule: sensory evaluation involving consumption of wholesome foods without additives.

**Informed Consent Statement:** Informed consent was obtained from all subjects involved in the study.

**Data Availability Statement:** Weather data are available through the NEWA website cited in the references. The fruit quality data are available upon request from the corresponding author. Farm identity remains confidential.

**Acknowledgments:** We thank Gavin Sacks and Gregory Peck for use of their lab and Richard Gaisser, David Zakalik, Michael Brown, Heather Scott and Terry Bates for their support in executing this research. Thanks to Gregory Peck and Courtney Weber for reviewing an early version of the manuscript.

**Conflicts of Interest:** The authors declare no conflict of interest.

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
