# Peer review of "Strawberry Flavor Is Influenced by the Air Temperature Differential during Fruit Development but Not Management Practices"

_agronomy, doi:10.3390/agronomy11030606_

Round 1

Reviewer 1 Report

Agronomy 1106996

The above paper examines the effect of temperature and plant agronomy on the performance of strawberries growing in New York State.  There were two studies, including a survey of several producers over three years, and a research experiment comparing organic and conventional agronomy at a single site over two years.  The results of the studies suggest that temperature has a major effect on fruit quality, whereas the impacts of plant agronomy are smaller.

I looked forward to reading a paper about an important issue affecting commercial berry production, namely fruit quality.  However, the paper has several flaws and is not suitable for publication at the moment.  The main issue with the study is that it lacks focus, and the reader is left wondering what factors contribute to poor yield/fruit quality in this growing area.  There is a general lack of statistical rigour.

The ‘Abstract’ is too general and needs to indicate a specific objective of the work.  It also needs to indicate the main findings of the study (regression between SCC, TA and temperature; impact of agronomic treatments on yield and quality).  Are there any recommendations for commercial producers?  Does conventional agronomy provide the best yields and returns?  Possible impact of global warming on fruit quality?

The ‘Introduction’ also suffers from too many general statements.  The authors need to provide specific information about the effect of organic and conventional agronomy on yield and fruit quality (SSC, TA, flavour).  Are there specific studies which indicate better yields/quality with organic agronomy?  I think the relationship between quality and N applications is also very inconsistent across studies.  Are there any meta-analyses on organic/conventional agronomy or nitrogen applications in strawberries?

The information on SSC and temperature is much better, but are there any studies relating yield/fruit TA/flavour to temperature?

The objective of the study needs to be tightened up.  Be specific.

The survey of the strawberry growers covered just a single cultivar.  The data relating to fruit SSC and TA seem to relate to just a single harvest for each farm.  There is also no mention of replicate samples for each farm and hence no measure of statistical/sampling variability.  No statistical comparison of the means across the different farms is possible (Figures 5 and 6).

Information is provided on the relationships between SSC and TA, and temperature across the different farms and years (Figures 1 to 3).  Some of the graphs include two regressions but the two different data sets are not identified.

The different temperature regimes used to explore the relationships between fruit quality and temperature are perplexing.  The authors use the lowest temperature between first flower and green fruit, or the difference between mean temperature and the lowest temperature from first flower to green fruit or from green to pink fruit.  The R2s for fruit SSC are rather low.  Those for fruit TA better.

I think it would be preferable to use other temperature regimes to explore these relationships, maybe 8 to 24 days before harvest.  MacKenzie used temperatures in the eight days before harvest in Florida.

The data in Figures 5 and 6 have no statistics to compare the means.  Maybe the best approach would be to indicate mean (with SE), median and maximum/minimum values across the study.

Figure 7 and Table 3 should be deleted.

The second part of the research examined the effect of agronomy on the performance of plants at a single site over two seasons.  The treatments were partially confounded with different N rates, N sources, organic matter supplements, microbial supplements and crop protection.  So the set-up of the experiment precluded the use of a three-way ANOVA to analyse the data.  At least there was significant replication, with 8 blocks per treatment.

Table 4 provides information on leaf nitrogen and carbon status, although the data appear to relate to only one sampling period.  Leaf N was higher with additional N in the LOC and CON treatments.  I failed to see how the treatments would affect leaf C status.  I think these data can be deleted.

Figure 8 provides data on yield for the field experiment, but the treatments don’t seem to be arranged in a logical order.  There is no statistical analysis (LSDs) to indicate if there are significant differences in the means of the various treatments.  The real interest is in marketable yield, with non-marketable yield secondary.  The authors indicate that the treatments have a similar percentage of marketable fruit (about 68%).

Most of the treatments appear to have similar yields, apart from CON+ which is higher.  LSDs need to be provided.

The important data on average fruit weight, SSC, TA and the flavour score from the panel over the two years need to be provided in a Table, along with the results of the ANOVA/F-test.

The data on fruit chemistry are secondary and should be deleted.  Same with the correlation analyses.

The ‘Discussion’ is very weak and needs to be completely rewritten.  The current version does not focus on the key findings of the study.  It needs to answer the question: does the weather and plant agronomy affect fruit quality, especially SCC and TA.  I think about a page or so should be sufficient to answer the question.

The ‘Conclusion’ is too long and does not clearly answer the question about fruit quality, weather and plant agronomy.  A single paragraph should be sufficient to outline the significance of the study.  Any recommendations.  What can growers do to improve fruit quality?  Does the response of SCC/TA to temperature have implications for strawberries under global warming?  Does conventional agronomy still provide the best commercial yields?

Reviewer 2 Report

This is an interesting study that explores the impacts of environmental variables and management practices on yield and fruit quality attributes of 'Jewel' matted row strawberry grown in the northeast United States. The approach of using samples from farms and a replicated experiment allows for unique analytical approaches to be combined in a single study to draw conclusions. The finding that weather impacted SSC and TA greatest and that sensory panels were unable to distinguish differences is important in that it demonstrates management practices (e.g. microbial dips and biostimulants) have no discernable effects and growers should be advised to focus on justified management practices that result in profitable outcomes while still maintaining agroecosystem health. 

I had minor comments in the methods looking for an explanation as to why harvest in 2020 was reduced, which was mentioned in the discussion. I feel an explanation should also be provided in the methods. Also, in the discussion more detail should be provided on the importance of timing of organic N fertilizer application and how results may be impacted with the selection of different cover crops. These are noted in the attached manuscript.  
